🔓 | **Open Peer Review** | Computational Biology | Research Article

# A comparison of short-read, HiFi long-read, and hybrid strategies for genome-resolved metagenomics

Raphael Eisenhofer,[1] Joseph Nesme,[2] Luisa Santos-Bay,[1] Adam Koziol,[1] Søren Johannes Sørensen,[2] Antton Alberdi,[1] Ostaizka Aizpurua[1]

**ABSTRACT** Shotgun metagenomics enables the reconstruction of complex microbial communities at a high level of detail. Such an approach can be conducted using both short-read and long-read sequencing data, as well as a combination of both. To assess the pros and cons of these different approaches, we used 22 fecal DNA extracts collected weekly for 11 weeks from two respective lab mice to study seven performance metrics over four combinations of sequencing depth and technology: (i) 20 Gbp of Illumina short-read data, (ii) 40 Gbp of short-read data, (iii) 20 Gbp of PacBio HiFi long-read data, and (iv) 40 Gbp of hybrid (20 Gbp of short-read +20 Gbp of long-read) data. No strategy was best for all metrics; instead, each one excelled across different metrics. The long-read approach yielded the best assembly statistics, with the highest N50 and lowest number of contigs. The 40 Gbp short-read approach yielded the highest number of refined bins. Finally, the hybrid approach yielded the longest assemblies and the highest mapping rate to the bacterial genomes. Our results suggest that while long-read sequencing significantly improves the quality of reconstructed bacterial genomes, it is more expensive and requires deeper sequencing than short-read approaches to recover a comparable amount of reconstructed genomes. The most optimal strategy is study-specific and depends on how researchers assess the trade-off between the quantity and quality of recovered genomes.

**IMPORTANCE** Mice are an important model organism for understanding the gut microbiome. When studying these gut microbiomes using DNA techniques, researchers can choose from technologies that use short or long DNA reads. In this study, we perform an extensive benchmark between short- and long-read DNA sequencing for studying mice gut microbiomes. We find that no one approach was best for all metrics and provide information that can help guide researchers in planning their experiments.

**KEYWORDS** microbiology, metagenomics, long read, mice, gut microbiome, microbiome

Shotgun metagenomics is increasingly becoming the preferred approach for characterizing the genetic composition of complex microbial communities, both in environmental and in host-associated samples (1). This is because, unlike targeted amplicon sequencing approaches, shotgun metagenomics enables the reconstruction of microbial genomes through metagenomic assembly and binning, making it a powerful tool (2–4). This process yields metagenome-assembled genomes (MAGs) (5), which can be annotated to provide an overview of the functional capabilities of each genome (6).

DNA sequencing technologies employed for generating shotgun metagenomic data can be grouped into two main categories (7). Short-read sequencing technology, which is the main strategy currently employed, yields large amounts of sequence data at a low cost. However, the short length of the generated sequences (usually up to 300 bp) can

**Ad Hoc Peer Reviewers** Antti Karkman, [1]Helsingin yliopisto, Helsinki, Finland; Longhao Jia, Institute of Science and Technology for Brain-Inspired Intelligence, Fudan University, Shanghai, China

Address correspondence to Raphael Eisenhofer, raphael.eisenhofer@sund.ku.dk.

The authors declare no conflict of interest.

See the funding table on p. 11.

10.1128/spectrum.03590-23 **1**

hinder the assembly of microbial genomes, as the underlying assembly graphs lack the information required to deal with repetitive and duplicated sequences (8). Long-read sequencing technologies can help overcome such limitations. While originally limited by high sequence error rates, novel strategies, such as the PacBio HiFi sequencing, enable the recovery of sequence qualities comparable to short reads (9). PacBio HiFi has recently been applied to various sample types, including human and sheep fecal samples (10, 11), chicken intestinal samples (12), anaerobic digesters (13), seawater (14), and saline lake sediments (15). However, the costs per data unit for long-read technologies are still several orders of magnitude higher than short-read technologies, which has hampered their widespread adoption. Hybrid assembly, which combines the strengths of both short- and long-read data, has been proposed as a cost-efficient strategy that enables taking advantage of the benefits of both approaches (16–18).

To the best of our knowledge, four other recent studies have compared the quality of MAG assembly from short-read to PacBio HiFi sequencing (15, 19–21). Tao et al. found that HiFi assembly provided the most complete bacterial and viral genomes and biosynthetic gene cluster (BGC), while a hybrid approach yielded a higher quantity and quality of MAGs (15). Zhang et al. and Orellana et al. also found that HiFi reads drastically improved the quality of bacterial MAGs but noted that more diversity was recovered using short reads (19, 21). In contrast, Gehrig et al. were able to recover both a higher quality and quantity of MAGs using HiFi reads over short reads (20). In this study, we compare the use of short read only, long read only, and hybrid sequencing strategies for recovering MAG catalogs from the fecal microbial communities of laboratory mice. We measured the performance by means of multiple metrics targeting quantitative and qualitative traits of the reconstructed MAG catalog. Based on our results, we discuss the suitability of using different sequencing strategies in light of study design, scientific aims, and available resources.

## MATERIALS AND METHODS

### Animal experiments and sample collection

The experiment was conducted at the ZIBA Experimentation Centre, Zarautz (Basque Country), between September and December 2020, under the animal experimentation licence PRO-AE-SS-154 issued by the Regional Government of Gipuzkoa. The mice sampled for this study were part of a larger project. Diversity Outbred mice of both sexes were acquired from the Jackson Laboratory at the age of 3 weeks, and the experiment was initiated after a 2-week acclimation period. The two study subjects (C04M3 = male and C13F5 = female) were housed separately along with four other animals in 840 cm$^2$ polycarbonate cages (Unno Type III, 38.2 × 22.0 cm) consisting of dust-free, 2–5-mm aspen bedding (ssniff, Germany), aspen nesting material, mixed sized chewing sticks, a small enrichment tube, and *ad libitum* access to water. Mice were fed once per day a total of 5 g of purified chow per animal. Cages were maintained in HPP750Life (Memmert, Germany) climate chambers that enabled full control and tracking of abiotic conditions. Simulated light cycles consisted of 10.5 hours of daylight and darkness and 3 hours of transitioning twilight time. Temperature was maintained at 24°C, humidity at 70%, and ventilation at 28 renovations/hour. Fecal samples were collected once a week for a period of 11 weeks. The animals were isolated in sterile cages for 30 minutes, and 80 mg of fecal material per animal was collected at each sampling point. Samples were immediately stored in 1 mL of DNA/RNA Shield presentation buffer (Zymo) and frozen at −20°C until DNA extraction.

### Data generation

Fecal samples (*n* = 22) were extracted using the in-house developed DREX protocol [details in Bozzi et al. (22)], preceded by a 10-minute bead-beating step at 30 Hz in 2-mL e-matrix tubes (MP Biomedical, USA) using a Tissuelyser II (Qiagen). The molarity and

fragment-length distribution of the extracts were measured using a Tapestation. Due to the DNA amount requirements and data generation costs, two different strategies were employed to generate short- and long-read sequencing data (Fig. 1). For short-read sequencing, 200 ng of each extract was fragmented using a Covaris LE220R focused ultrasonicator to 320–420 bp-long DNA fragments, and 11 individual sequencing libraries per animal were prepared using the BEST protocol (23). DNA sequencing was performed on an Illumina NovaSeq 6000 platform, using S4 150 paired-end chemistry, and aiming for ca. 4 Gbp (1 Gbp = $1 \times 10^9$ bp) of data per library (ca. 40 Gbp per animal). For long-read sequencing, the 11 DNA extracts (145.5 ng each) of each individual were pooled to prepare a single library. Fragmented DNA (~7,000 bp) was prepared for PacBio sequencing using the SMRTbell express template prep kit 2.0 (Pacific Bioscience). SMRTbell libraries were bound with sequencing primer v5 and Sequel II DNA Polymerase 2.0 using Sequel II Binding Kit 2.2. Bound complexes were sequenced on a PacBio Sequel IIe platform using Sequencing Reagents plate 2.0. PacBio Circular Consensus Sequence (CCS) reads are produced by calling consensus of subreads of a single DNA molecule ligated with hairpin adapters, resulting in a circularized DNA molecule and allowing several passes of DNA polymerase on each strand. Circular consensus reads were generated onboard, and only HiFi reads (Phred score >Q20) were used for downstream analysis. Each individual pool yielded approximately 20 Gbp of HiFi reads per library, with an average read size of around 7 kbp.

## Data analysis

Raw data generated from Illumina ($n = 22$) and PacBio ($n = 2$) were processed following four contrasting strategies: SR20) short-read assembly from 20 Gbp/mouse of data, SR40) short-read assembly from 40 Gbp/mouse of data, LR) long-read assembly 20 Gbp/mouse, and HY) hybrid assembly [20 Gbp of short reads +20 Gbp of long reads (LRs) per individual]. Raw short-read sequencing data were preprocessed using the Earth Hologenome Initiative (EHI) preprocessing snakemake pipeline (https://github.com/earthhologenome/EHI_bioinformatics), briefly: Fastp v0.23.1 (24) was used to trim the adapters and low-quality bases, followed by mapping using Bowtie2 v2.4.4 (25) and samtools v1.12 (26) to the *Mus musculus* reference genome (GRCm39) to remove

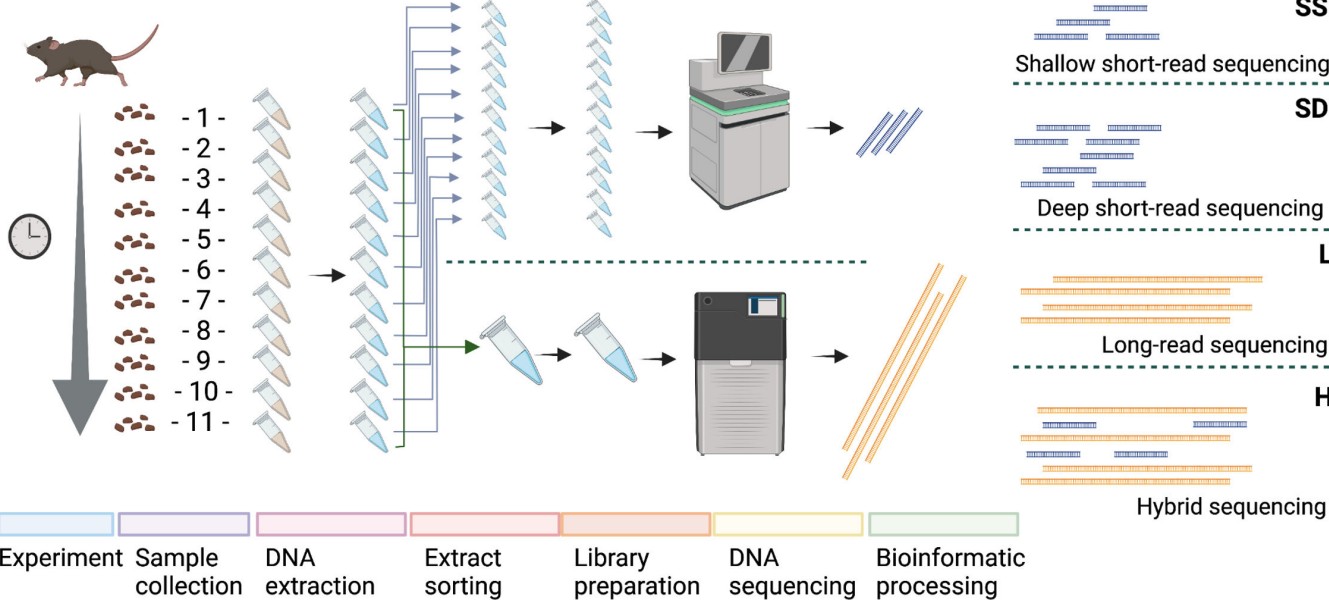

**FIG 1** Overview of the study design. Fecal samples ($n = 22$) were collected from two mice across a longitudinal study and processed using two different laboratory protocols to generate short-read and long-read sequencing data, resulting in four datasets compared in the study. SS = short-read 20 Gbp, SD = short-read 40 Gbp, L = long-read 20 Gbp, and H = hybrid assembly 40 Gbp. Created with BioRender.com

host reads. Minimap2 (27) was used for long-read mapping to the *M. musculus* reference genome. The filtered reads were then randomly subsampled using seqtk v1.3 (seed = 1337) (https://github.com/lh3/seqtk).

Short-read co-assemblies (SR20 and SR40) were constructed using the EHI snakemake pipeline (https://github.com/earthhologenome/EHI_bioinformatics), briefly: metaSpades v3.15.3 was used for assembly (-k 21,33,55,77,99), followed by the MetaWRAP (28) binning and refinement pipelines, which use MaxBin2 (29), MetaBAT2 (30), CONCOCT (31), and CheckM (32). For the hybrid assembly (HY), metaSpades v3.15.3 (33) was used with the --pacbio flag. LRs were assembled using hifiasm-meta r63 (34), and binning/refinement was performed using the Pacific Biosciences HiFi-MAG-Pipeline (https://github.com/PacificBiosciences/pb-metagenomics-tools) (release 2.0.2), which uses minimap2 for mapping long reads to assemblies (27), CheckM 2 (35), semibin (36), and DAS_tool (37). Note that long-read assemblies were binned individually. QUAST v5.2.0 (38) was used to generate assembly metrics, and CoverM 0.6.1 (https://github.com/wwood/CoverM) was used to calculate the percentage of reads mapping to co-assemblies. GTDB-tk (v2.1.0; database = R207_v2) was used to taxonomically annotate MAGs (39, 40).

MAG dereplication was performed using the EHI snakemake pipeline (https://github.com/earthhologenome/EHI_bioinformatics), briefly: dRep v3.4.0 (41) was run at 98% ANI (average nucleotide identity). Dereplicated MAGs were then concatenated and indexed before being used as a reference for Bowtie2 mapping. Final count tables were generated with CoverM. antiSMASH v6.1.1 (42) was used to annotate biosynthetic gene clusters, and Barrnap v0.9 (https://github.com/tseemann/barrnap) was used to annotate 5S, 16S, and 23S rRNA genes. tRNAs were annotated using tRNAscan-SE 2.0 (43). For each strategy, we analyzed the seven metrics defined in Table 1.

Complete circular sequences were identified from metagenome assembly graphs for each data set based on circularity using Sequence Contents-Aware Plasmid Peeler (SCAPP) (44) and annotated as plasmid, viral or chromosomal sequences based on specific marker genes using viralVerify v1.1 (45). Short and long reads were mapped to respective assemblies using minimap2 v2.17 (27) using "-ax sr" and "-ax map-pb" options, respectively, for short and long reads. Short and long reads mapped to Hybrid 40 Gbp assembly were merged before the SCAPP step. Circular contigs from metagenomes assembly graphs were extracted using SCAPP (44) using each data set assembly and respective mapped reads. Circular contigs were classified as plasmid, viral, or chromosomal sequences using viralVerify v1.1 (45).

Summary statistics and visualizations were performed in R using the tidyverse (46) and phyloseq (47) packages. For code specifics, see the github repository (https://

**TABLE 1** Description of the seven metrics considered for assessing the performance of the different sequencing strategies

| Metric | Description |
| --- | --- |
| Assembly length | The total number of nucleotides of the unfiltered (no minimum contig size threshold applied) metagenomic assembly. |
| Assembly contiguity | The N50 metric of assembly contiguity, defined as the length of the shortest contig for which longer and equal-length contigs cover at least 50% of the assembly. |
| Assembly coverage | Percentage of short-read sequences (a random subset of 20 Gbp) mapping to the assembly. |
| MAG number | Number of metagenome-assembled genomes with >70% CheckM completeness and <10% redundancy recovered from the sequencing data. |
| MAG contiguity | Average number of contigs per MAG. |
| MAG completeness | Average completeness of the MAGs in terms of CheckM assessment based on the presence of single-copy core genes. |
| MAG contamination | Inverse to the average contamination of the MAGs in terms of CheckM assessment based on the redundancy of single-copy core genes. |

github.com/EisenRa/2023_HiFi_comparison_mice). Raw sequencing reads are available at ENA accession: PRJEB65885.

## Cost estimations

Cost estimations were carried out by averaging prices obtained in early 2023 from multiple commercial sequencing service providers for DNA extraction, library preparation, and sequencing short- and long-read sequencing data (Table 2).

## RESULTS AND DISCUSSION

The four assembly strategies compared in this study (Fig. 1) were built from a total of 78.6 Gbp of short-read and 39.6 Gbp of long-read sequencing data generated over one NovaSeq 6000S4 and two PacBio Sequel IIe HiFi sequencing runs, respectively. Both sequencing strategies yielded DNA reads with a comparable level of quality (>40 Phred score, indicating a probability for erroneous base calling below 0.001). However, the read lengths differed significantly: the short-read sequencing yielded consistent 150-bp long-paired reads, while the long-read sequencing generated single reads with a mean length of 6,792 bp (6,351 bp for C04M3 and 7,233 bp for C13F5).

### Assembly contiguity, length, and coverage

The long-read approach yielded metagenomic assemblies with an average assembly contiguity of 9–18 times higher than short-read and hybrid approaches (Table 3; Fig. 2 and 3). While the N50 value for short-read metagenomic assemblies (after filtering out <1,500 bp contigs) was around 20 kbp (20,000 bases), the N50 values of long-read assemblies were between over 235 and 519 kbp. This result is not surprising, since long-read sequences are known to improve assembly contiguity (48). The hybrid assembly exhibited a noteworthy increase in contiguity with respect to short-read only approaches (ca. 40 kbp), but it was far lower than the values of long-read assemblies. This is likely due to the fact that hybridSPAdes constructs the initial assembly graph with short reads, using the long reads to close the gaps and resolve repeats (33). Long-read assembly also yielded the longest assembly length (Table 3). The hybrid assembly approach recruited the most sequences through read-mapping (96% and 97% of reads); thus, exhibiting the highest assembly coverage across all strategies. Yet, all approaches managed to capture >93% of the metagenomic reads, indicating that most of the complexity in these samples was captured.

### Quantity of recovered MAGs

The four strategies yielded a total of 619 redundant MAGs, which resulted in 125 non-redundant MAGs after dereplication at 98% ANI (Table 4; Fig. 2). Short-read approaches yielded a higher number of MAGs than long-read-based strategies (Fig. 3). For the same amount of nucleotide data generated (20 Gbp), the short-read approach resulted in 10 more dereplicated MAGs (at 98% ANI) compared to the long-read approach. The analysis of MAG-recovery through different strategies showed that the deficit of MAGs in the long-read approach was mainly due to the inability to recover less abundant bacteria (Fig. 4). The MAGs missed by the long-read approach accounted for on average 9% of the mapped reads. The hybrid approach recruited the highest number of reads (93% and 94%) into the dereplicated MAG catalog. In contrast, the long-read approach recruited the lowest number of reads (62% and 70%). While this may seem surprising given that the long-read assemblies were the largest, it is possible that these

**TABLE 2** Approximate cost estimations (as of mid-2022) for generating short-read (Illumina NovaSeq 6000S4) and long-read (PacBio Sequel IIe HiFi) sequencing data based on average quotes of multiple sequencing service providers

| | Cost per DNA extraction | Cost per library preparation | Sequencing cost per GB | Total cost of 20 GB of data |
|---|---|---|---|---|
| Short-read sequencing | 20 USD | 50 USD | 10 USD | 270 USD |
| Long-read sequencing | 20 USD | 350 USD | 200 USD | 4,370 USD |

**TABLE 3** Assembly statistics[a]

| Sample | Assembly length (Mbp) | Assembly contiguity (N50 kbp) | Assembly coverage (short reads mapped) (%) |
|---|---|---|---|
| C04M3_Short_read_20Gbp | 345 | 20 | 93 |
| C13F5_Short_read_20Gbp | 295 | 20 | 94 |
| C04M3_Short_read_40Gbp | 451 | 30 | 95 |
| C13F5_Short_read_40Gbp | 339 | 24 | 96 |
| C04M3_Hybrid_40Gbp | 443 | 39 | **96** |
| C13F5_Hybrid_40Gbp | 387 | 41 | **97** |
| C04M3_Long_read_20Gbp | **505** | **236** | 94 |
| C13F5_Long_read_20Gbp | **503** | **520** | 96 |

[a]Bold indicates highest values for a given measure.

assemblies were more redundant than the short-read assemblies. Orellana et al. (19) also reported that a greater diversity of MAGs was obtained using short reads compared to HiFi reads, which they concluded was likely due to short reads having higher sequencing depth in their study. In this study, we controlled for sequencing depth in our comparison and still found that HiFi reads yielded fewer MAGs.

Large MAG catalogs are publically available for some well-studied host species and environments, such as humans and mice. We wanted to compare the proportion of reads that could be recruited to the Mouse Gastrointestinal Bacteria Catalog (MGBC) (49) versus *de novo* MAGs created in the present study. Overall, 26,660 MAGs from the MGBC were dereplicated at 98% ANI into 1,363 MAGs, which were used as reference for the mapping. The MGBC recruited on average 86% of reads, similar to what was obtained

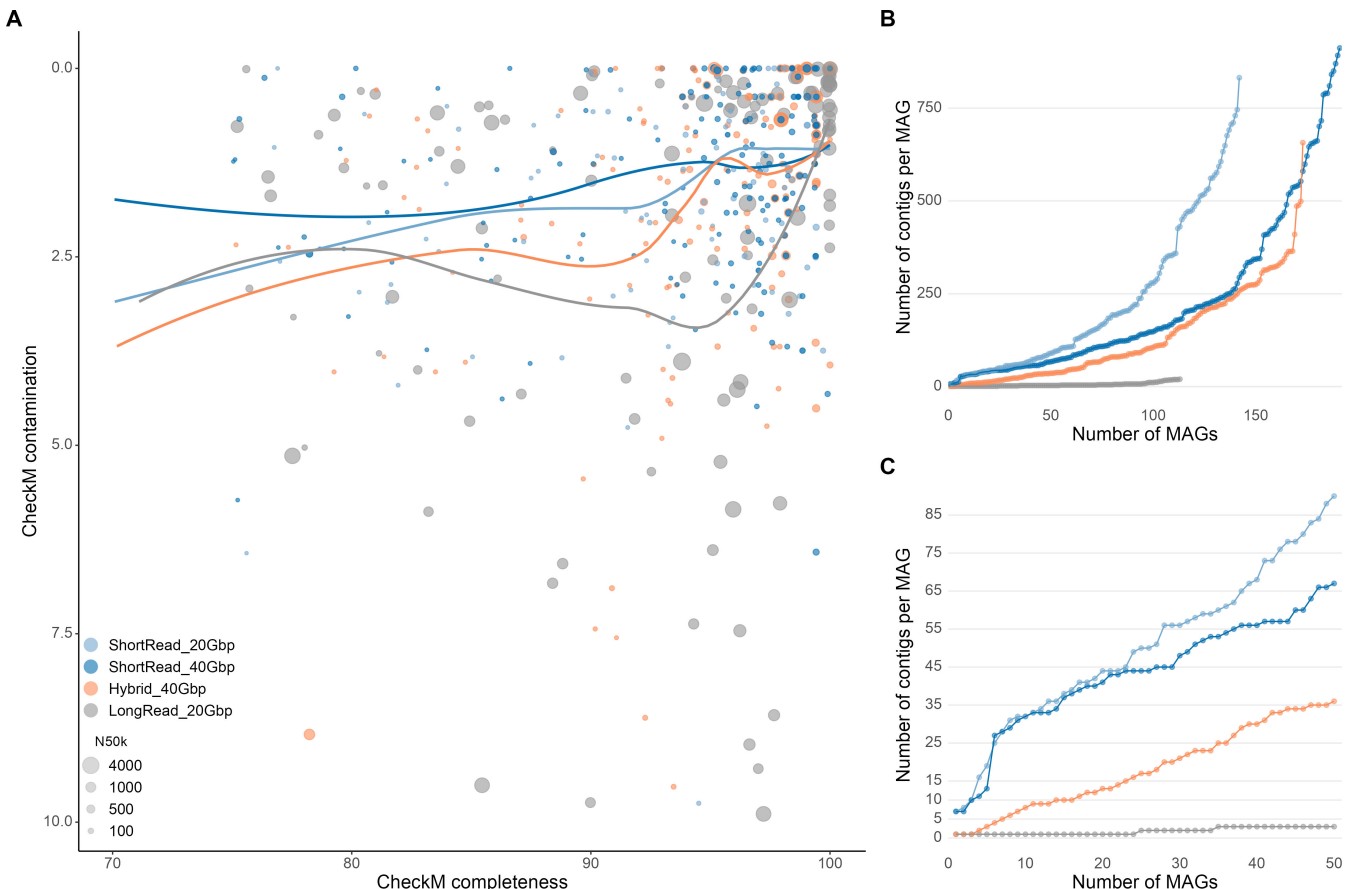

**FIG 2** (A) Statistics of the assemblies and MAGs recovered through the different strategies. (B) All MAGs, (C) top 50 most contiguous MAGs.

**TABLE 4** MAG statistics[a]

| Sample | MAG coverage (short reads mapped) | MAG number | MAG number dereplicated | MAG contiguity (mean # of contigs) | MAG completeness (%) | MAG contamination (%) | Circularized MAGs |
|---|---|---|---|---|---|---|---|
| C04M3_Short_read_20Gbp | 87% | 75 | 47 | 230 | 92.7 | 1.6 | 0 |
| C13F5_Short_read_20Gbp | 87% | 67 | 46 | 225 | 94.8 | 1.3 | 0 |
| C04M3_Short_read_40Gbp | 87% | **108** | 80 | 212 | 92.4 | 1.4 | 0 |
| C13F5_Short_read_40Gbp | 89% | **83** | 49 | 244 | 93.3 | 1.4 | 0 |
| C04M3_Hybrid_40Gbp | **93%** | 90 | 62 | 127 | 93.3 | 1.8 | 1 |
| C13F5_Hybrid_40Gbp | **94%** | 83 | 52 | 135 | 93.6 | 1.8 | 2 |
| C04M3_Long_read_20Gbp | 70% | 52 | 35 | **5** | 93.3 | 2.3 | **11** |
| C13F5_Long_read_20Gbp | 62% | 61 | 48 | **4** | 90.5 | 2.4 | **10** |

[a]Bold indicates highest values for a given measure.

with our *de novo* short-read catalogs (87%–89%), but less than the *de novo* hybrid catalog (93%–94%) (Table 4). While publically available MAG catalogs can be useful for comparative analyses, we recommend that *de novo* catalogs be created for a given study where possible (if sequencing depth is sufficient), as study-specific genes or strains may be absent from large publicly available MAG catalogs.

## Quality of recovered MAGs

Long-read sequencing outperformed the rest of the strategies in terms of MAG quality (Fig. 3). MAG contiguity was significantly better for long-read sequencing-based MAGs, with an average of 5 contigs per MAG versus an average of >200 contigs for MAGs reconstructed from short-read data (Table 4; Fig. 2). There was no major difference in MAG completeness among the assembly strategies (Table 4; Fig. S1). The long-read strategy successfully recovered 21 circularized genomes (14 dereplicated at 98%), of which only two had CheckM2 completeness scores of 100%, highlighting that genome completion estimation methods come with a level of uncertainty and are not perfect (50). MAG contamination scores were generally low (lower scores indicate less contamination) and similar for all strategies, with the long-read approach exhibiting slightly worse values (Table 4; Fig. S1). Beyond CheckM2 completeness and contamination estimates, we also compared how long reads affected the number of high-quality MAGs obtained as defined by the MIMAG criteria (CheckM completeness >90%, contamination <5%, 5S, 16S, 23S rRNA genes present, at least 18 tRNA genes present) (51): 44.2% of long-read MAGs were high-quality (50/113), 23.1% for hybrid MAGs (40/173), 3.5% for short-read 20-Gbp MAGs (5/142), and 3.1% for short-read 40-Gbp MAGs (6/191).

Secondary metabolite BGCs can be difficult to assemble using short reads (52). We used antiSMASH to compare how many BGCs were reconstructed in each of the assembly approaches. While the short-read and hybrid approaches yielded the highest number of BGCs (334), when normalized by the number of MAGs in each approach, long reads performed similarly (Table 5). Additionally, long-read assembly reconstructed over twice as many complete BGCs and substantially longer BGCs compared to the other approaches (Table 5). We did not observe marked differences in the diversity of BGCs between the different assembly approaches (Fig. S2). Finally, it is well known that 16S rRNA genes are particularly difficult to assemble and bin using short reads (53). We found

**TABLE 5** Secondary metabolite BGC results from anti-SMASH[a]

| Assembly | Number of BGCs | BGCs/MAGs | Complete BGCs (%) | Mean BGC length (bp) |
|---|---|---|---|---|
| Short_read_20Gbp | 262 | 1.85 | 36.3 | 17,511 |
| Short_read_40Gbp | **334** | 1.75 | 38.6 | 18,274 |
| Hybrid | **334** | 1.93 | 45.2 | 20,213 |
| Long_read | 219 | **1.94** | **96.3** | **25,268** |

[a]Bold indicates highest values for a given measure.

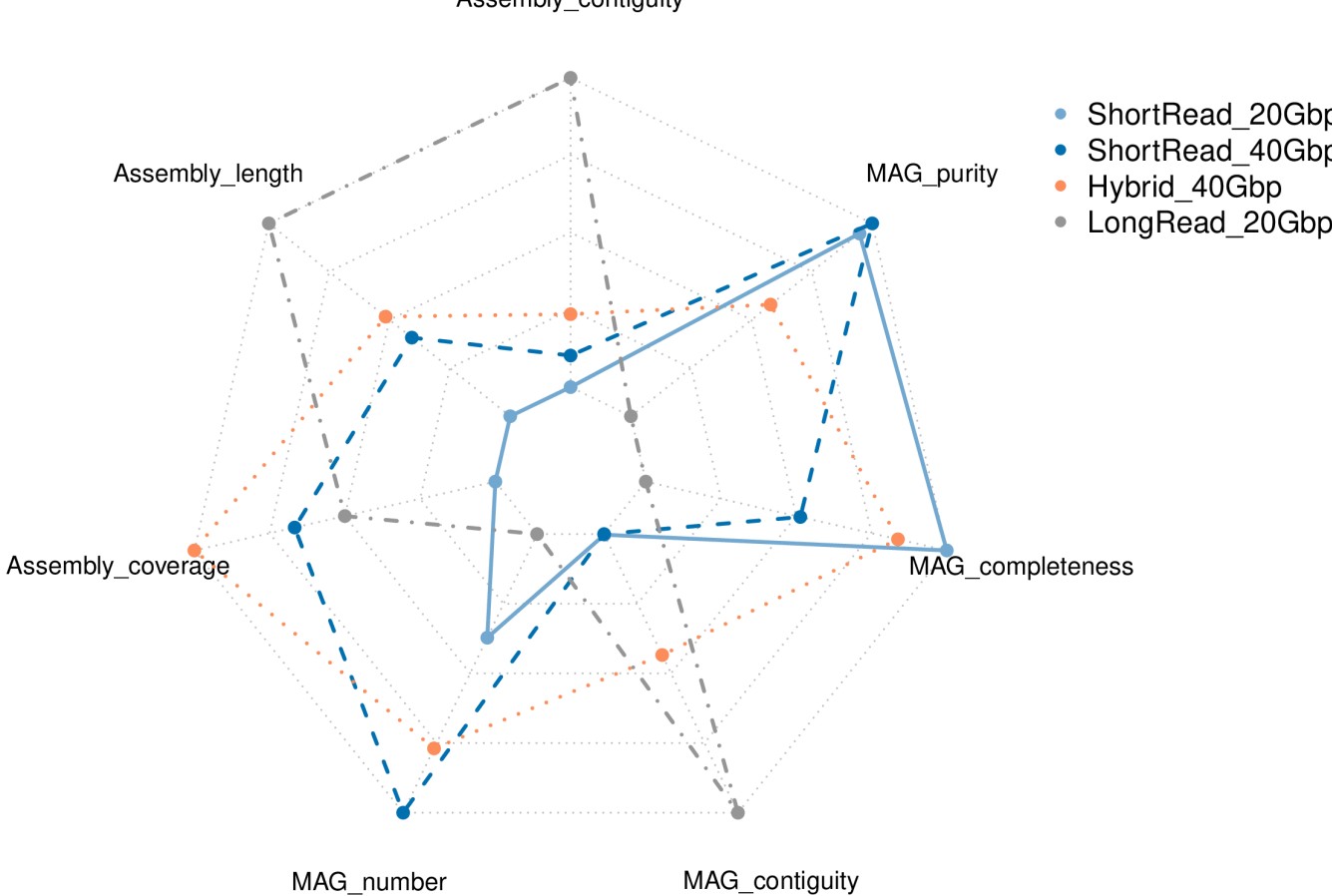

**FIG 3** Overview of the seven metrics employed for assessing the performance of the different approaches for genome-resolved metagenomics.

that long-read assembly substantially improved the reconstruction of 16S rRNA genes compared to short reads (18 versus 365) (Fig. S3).

## Plasmids, viruses, and assembled circular elements

Plasmids, viruses, and other mobile genetic elements (MGEs) are of particular biological interest both in terms of public health and microbial adaptation, considering that the vast majority of antibiotic resistance acquisitions in pathogenic bacteria are mediated by plasmids (54) and that viruses are an important, albeit overlooked, component of the gut microbiome (55). Plasmids are also central vectors for the genetic exchange between bacterial chromosomes (56). However, the study of plasmids in the past decade has been greatly hampered due to the challenges associated with plasmid assembly using short-read sequencing (57). A definite advantage of long- in contrast to short reads is the ability to span repeats of a few kilobases within a single sequencing read and to capture entire circular plasmids within a single read—increasing confidence that the assembled plasmids are not chimeric. To test the advantage of long- over short reads for plasmid metagenomic, we extracted circular sequences from each metagenome assembly graph and assigned these contigs to plasmid, viral, or chromosomal sequences. We counted the number of and total length of circularized sequences obtained from each type of data sets (Fig. 5; Table 6). The number of circularized contigs varies significantly between short- and long-read strategies. For the two short-read and hybrid strategies, between 1,510 and 1,980 circular contigs were identified (Table 6), while only 123 were recovered for the long-read-only approach. Despite the order of magnitude fewer circular contigs were obtained for long reads, the total length of these reached 42.4 Mbp, while

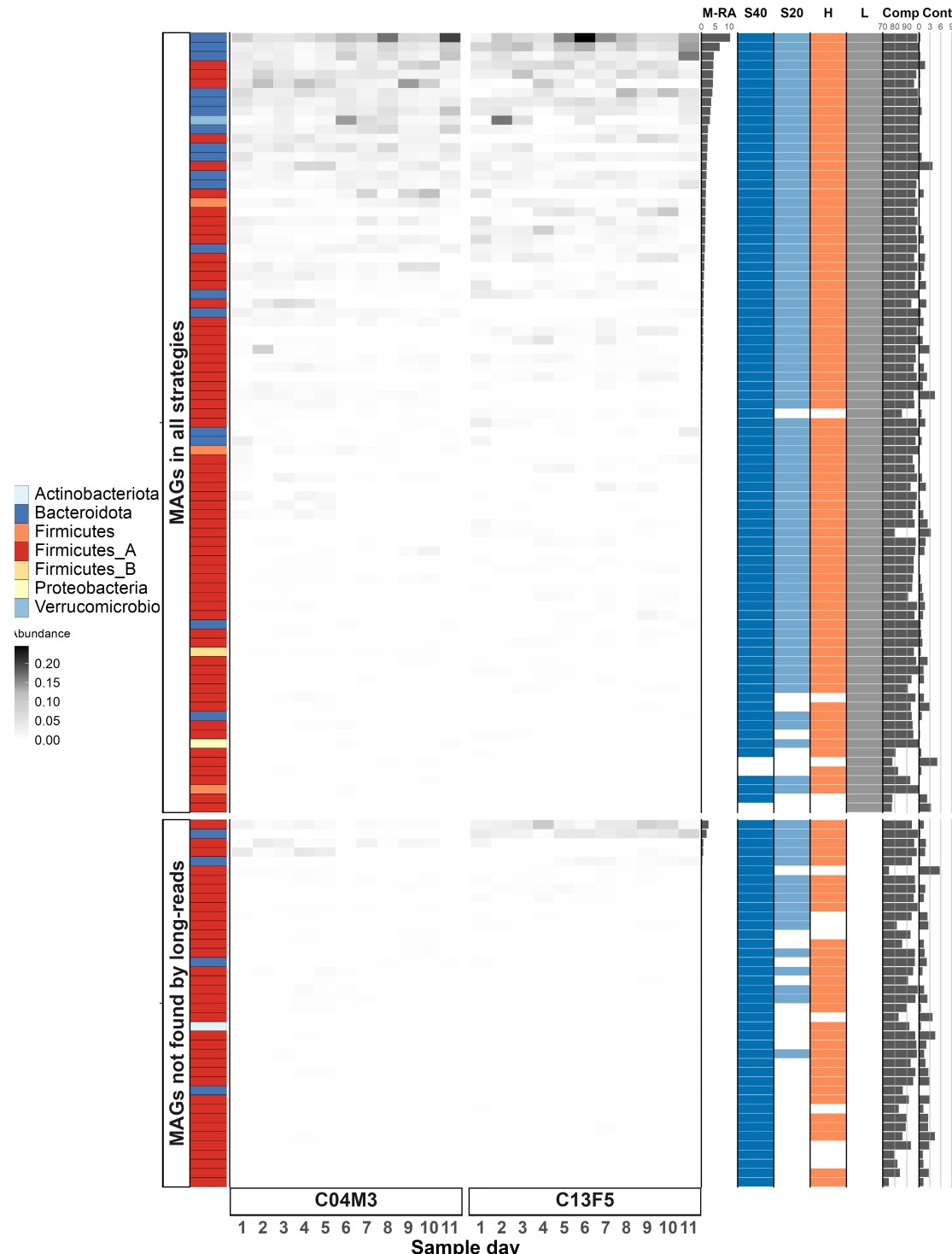

**FIG 4** Overview of MAG reconstruction performance by the different strategies (S40 = 40 Gbp short-read, S20 = 20 Gbp short-read, H = hybrid, L = HiFi long-reads). All MAGs (619) were dereplicated at 98%. Each MAG is a row and each column is a sample. Filled cells indicate the relative abundance of reads (20 Gbp short-reads) mapping to a given MAG. The MAGs are faceted by whether or not they were assembled using long reads. M-RA = mean relative abundance of reads mapping to a given MAG. The four colored columns indicate whether a given assembly strategy reconstructed a MAG. Comp and Cont = CheckM estimates for MAG completeness and contamination, respectively.

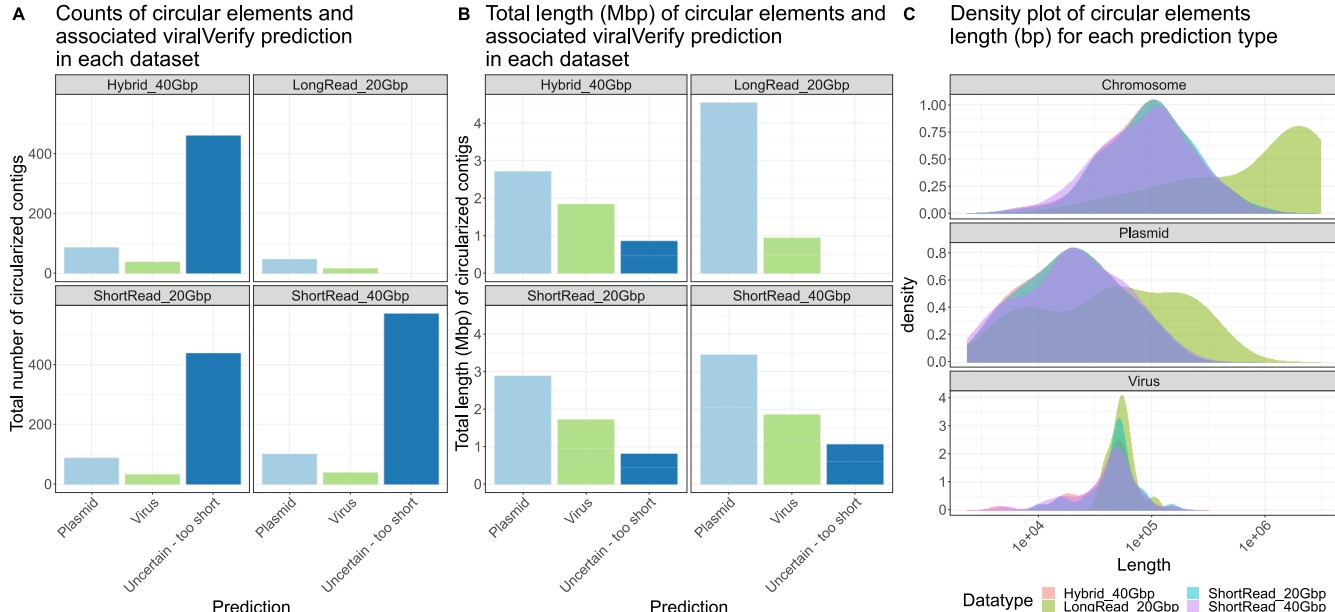

**FIG 5** Summary of circular contig extraction from each data set type annotated as plasmid, virus, or too short to be confidently assigned. (A) Total number of circular contigs identified and annotated by viralVerify. (B) Total length in megabase pairs (Mbp) of circular contigs. (C) Circular contig length density for chromosome, plasmid, and virus prediction by viralVerify. The counts and total length of all predictions are displayed in Fig. S3.

short-read and hybrid approaches yielded between 71 Mbp and 89.2 Mbp, indicating that circular sequences obtained by long-read assemblies are significantly larger than with the short-read ones.

A number of genetic structures on chromosomes contain repeats, for example, transposons and insertion sequences possess inverted repeats at their ends (58). Mobile genetic elements such as integrons or transposons will make circular double-stranded DNA intermediates when excised from the chromosome during transposition and can be part of the total extracted DNA pool. Repeats and circular chromosomal structures will produce a circular path in contig assembly graphs and will be extracted as circular contigs by SCAPP. In addition, many dsDNA viruses have circular genomes. Circular contigs were therefore classified using marker genes specific for chromosomal, plasmid, or viral sequences with viralVerify (45) (Fig. 5; Fig. S4). Most circular contigs obtained with short-read and hybrid approaches were too short to be classified as chromosomal or plasmids sequences (Fig. 5A; Fig. S4A). In contrast, the majority of circular sequences from the long-read approach were annotated as plasmids, with no sequences too short for classification. When comparing the predictions of replicon type, all approaches showed that chromosomal contigs account for the majority (84.43%–85.87%) of the total circular contigs length (Table 6). Both short-read and hybrid approaches yielded between 9.6% and 8.6% of uncertain sequences and only between 4% and 2.9% of plasmid sequences for the long-read approach, only 2% of the contigs could not be assigned and up to 10.7% of plasmid circular sequences were assembled from 20 Gb of PacBio data.

**TABLE 6** Plasmids and circular contigs statistics[b]

| Assembly | Number of circular contigs | Total length of circular contigs (bp) | Plasmid[a] total length (bp) | Virus[a] total length (bp) | Chromosome[a] total length (bp) | Uncertain[a] total length (bp) |
|---|---|---|---|---|---|---|
| Short_read_20Gbp | 1510 | 7.10 + 07 | 2.89E + 06 (4.07%) | **1.73E + 06 (2.43%)** | 6.07E + 07 (85.48%) | 5.7E + 06 (8.02%) |
| Short_read_40Gbp | **1980** | **8.92E + 07** | 3.45E + 06 (3.87%) | 1.86E + 06 (2.09%) | 7.53E + 07 (84.43%) | 8.57E + 06 (9.61%) |
| Hybrid | 1624 | 7.77E + 07 | 2.72E + 06 (3.5%) | 1.85E + 06 (2.38%) | **6.67E + 07 (85.87%)** | 6.41E + 06 (8.26%) |
| Long_read | 123 | 4.24E + 07 | **4.54E + 06 (10.72%)** | 9.52E + 05 (2.25%) | 3.6E + 07 (84.97%) | **8.75E + 05 (2.07%)** |

[a]based on viralVerify prediction, see methods.
[b]Bold indicates highest values for a given measure.

This indicates that circular sequences obtained by the long-read approach were easier to classify and recovered more larger plasmid sequences above 100 kb which were not recovered in short-read data sets (Fig. 5C).

Short- and long-read approaches performed equally well in recovering circular viral genomes. All methods were in agreement regarding their size distribution—centered between 25kbp and 75kbp (Fig. 5C). This is in accordance with known sizes of DNA viruses infecting bacteria and archaea that are likely the most abundant hosts in the gut microbiome obtained from fecal samples (59–61). This suggests that if viruses are the targeted diversity for a study, the addition of long reads may not substantially improve the results. In contrast, if plasmids are of focus, long reads can recover sizes that fit the broad distribution of known plasmids from databases (62). However, larger-size plasmids (>100 kb) are notably absent from short-read data sets and could only be found with the long-read approach in the same samples. This is an important consideration when investigating the microbial ecology of plasmids where large conjugative plasmids play a prominent role (63).

## Conclusions

In conclusion, our study compared the performance of different sequencing approaches for shotgun metagenomics using fecal DNA extracts from lab mice. By assessing seven performance metrics across four combinations of sequencing depth and technology, we gained valuable insights into the strengths and limitations of the individual approaches. Our results reveal a trade-off between the use of short-read and long-read sequencing for genome-resolved metagenomics. While deep long-read sequencing produced the highest-quality assemblies, the associated costs remain prohibitive. The optimal sequencing strategy depends on the specific goals and priorities of the study. Researchers need to carefully evaluate the balance between the quantity and quality of recovered genomes, and extrachromosomal elements taking into account available resources and project requirements. There is no one-size-fits-all approach, and a thoughtful and tailored selection of sequencing strategies is essential. Additionally, our study contributes new HiFi long-read sequencing data for the extensively studied laboratory mouse, which will serve as a valuable resource for future benchmarks and bioinformatic developments.

## AUTHOR AFFILIATIONS

[1]Center for Evolutionary Hologenomics, Globe Institute, University of Copenhagen, Copenhagen, Denmark
[2]Section of Microbiology, Department of Biology, University of Copenhagen, Copenhagen, Denmark

## AUTHOR ORCIDs

Raphael Eisenhofer ⓘ http://orcid.org/0000-0002-3843-0749
Søren Johannes Sørensen ⓘ http://orcid.org/0000-0001-6227-9906
Antton Alberdi ⓘ http://orcid.org/0000-0002-2875-6446
Ostaizka Aizpurua ⓘ http://orcid.org/0000-0001-8053-3672

## FUNDING

| Funder | Grant(s) | Author(s) |
| --- | --- | --- |
| Danish National Research Foundation award | DNRF143 | Antton Alberdi |
| | | Ostaizka Aizpurua |
| The Carlsberg Foundation | CF20-0460 | Antton Alberdi |

## AUTHOR CONTRIBUTIONS

Raphael Eisenhofer, Conceptualization, Data curation, Formal analysis, Investigation, Methodology, Software, Visualization, Writing – original draft, Writing – review and editing | Joseph Nesme, Formal analysis, Investigation, Methodology, Visualization, Writing – review and editing | Luisa Santos-Bay, Investigation, Methodology, Writing – review and editing | Adam Koziol, Investigation, Methodology, Writing – review and editing | Søren Johannes Sørensen, Funding acquisition, Project administration, Supervision, Writing – review and editing | Antton Alberdi, Conceptualization, Formal analysis, Funding acquisition, Investigation, Project administration, Supervision, Visualization, Writing – original draft, Writing – review and editing | Ostaizka Aizpurua, Conceptualization, Formal analysis, Funding acquisition, Investigation, Project administration, Supervision, Visualization, Writing – review and editing

## DATA AVAILABILITY

For code specifics, see the github repository (https://github.com/EisenRa/2023_HiFi_comparison_mice). Raw sequencing reads are available at Bioproject: PRJEB65885

## ETHICS APPROVAL

The animal experiment was approved by the Regional Government of Gipuzkoa under experimentation licence PRO-AE-SS-154.

## ADDITIONAL FILES

The following material is available online.

### Supplemental Material

**Figure S1 (Spectrum03590-23-s0001.tiff).** CheckM completeness and contamination scores for the metagenome assembled genomes.
**Figure S2 (Spectrum03590-23-s0002.tiff).** Counts of different biosynthetic gene clusters predicted by antiSMASH.
**Figure S3 (Spectrum03590-23-s0003.tiff).** Number of 16S rRNA genes recovered by different assembly types.
**Figure S4 (Spectrum03590-23-s0004.eps).** Summary of circular contigs extraction from each data set type annotated by viralVerify.
**Legends (Spectrum03590-23-s0005.docx).** Legends for the supplemental figures.

### Open Peer Review

**PEER REVIEW HISTORY (review-history.pdf).** An accounting of the reviewer comments and feedback.

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
