## [Reviewer comments · Microbiology Spectrum]

Microbiology Spectrum

A comparison of short-read, HiFi long-read, and hybrid strategies for genome-resolved metagenomics

Raphael Eisenhofer, Joseph Nesme, Luisa Santos-Bay, Adam Koziol, Søren Sørensen, Antton Alberdi, and Ostaizka Aizpurua

Corresponding Author(s): Raphael Eisenhofer, Kobenhavns Universitet

Review Timeline:

Submission Date:	October 7, 2023
Editorial Decision:	December 11, 2023
Revision Received:	January 24, 2024
Accepted:	February 11, 2024

Editor: Wei-Hua Chen

Reviewer(s): Disclosure of reviewer identity is with reference to reviewer comments included in decision letter(s). The following individuals involved in review of your submission have agreed to reveal their identity: Katie Sipes (Reviewer #1); Antti Karkman (Reviewer #2); Longhao Jia (Reviewer #3)

Transaction Report:

DOI: <https://doi.org/10.1128/spectrum.03590-23>

Re: Spectrum03590-23 (A comparison of short-read, HiFi long-read, and hybrid strategies for genome-resolved metagenomics)

Dear Dr. Raphael Eisenhofer:

Thank you for the privilege of reviewing your work. Your manuscript has been evaluated by three external experts in the field. As you will see below, although they all found your research interesting, they all raised major concerns. Please address those concerns carefully, revise your manuscript, and provide a point-to-point response. Your revised manuscript will be evaluated again by the same experts.

Revision Guidelines

Sincerely,
Wei-Hua Chen
Editor
Microbiology Spectrum

Reviewer #1 (Comments for the Author):

Review: A comparison of short-read, HiFi long-read, and hybrid strategies for genome-resolved metagenomics

Summary:

The authors compared seven performance metrics over four sequencing combinations from a laboratory mouse study. While different strategies were better at different aspect, the long-read gave the best assembly statistics and the hybrid approach gave the longest assemblies and highest percent of reads mapped back to genomes. This study thoughtfully explores the pros and cons of each of the four sequencing depth and technologies.

Minor Comments:

Line 127: "Binning and refinement were performed as above." Is not needed.

Line 157: There were no record found under ENA accession: PRJEB65885

Major Comments:

There are no major comments; this has been a long-awaited comparison.

Reviewer #2 (Comments for the Author):

In the submitted manuscript Eisenhofer et al. compare different sequencing strategies in the light of genome-resolved metagenomics. The manuscript is very well written and the results are clearly presented. I have to say that I really enjoyed reading this manuscript and I think the output from this manuscript is important to the field. Of course I have some comments and criticisms and they are listed below.

1. My main concern is related to binning. It's not clearly written (and it should be) how the original samples have been mapped back to the co-assemblies. I would assume that in the case of short-reads the authors have used all samples separately and mapped them. In long-read approach the original samples were pooled before sequencing, so there is only one (?) sample mapped back to the assembly. As most of the binning algorithms rely on nucleotide composition and differential coverage, the information from a single sample mapping is very weak compared to differential coverage coming from 11 samples. So if this was the case, the differences seeing in the recovery of MAGs are actually stemming from the binning, not really from the sequencing strategy. I hope the authors would clarify this part and in case there was really different strategy in the binning between the approaches, this should be fixed to make the comparison really about sequencing strategy. And of course the binning software were totally different for the two short- and long-read approaches, I would assume this also affects the results. I don't think there would be any issues using the same approach for all strategies, but correct me if I'm wrong.

2. As this is a very methods oriented paper, the methods could be written with more details. It is not clear whether the authors used a different mapping strategy when mapping the short- and long- reads back to the assembly. Or did they use bowtie2 for both approaches when removing the host sequences? I think bowtie2 does not work well with long-reads. Again, correct me if I'm wrong.

3. I would like to see some discussion on why the long-read approach yielded the largest assembly, but had a lower mapping rate compared to the other approached. I would expect that a bigger assembly would also cover more of the original data.

4. Line 204: CheckM2 does not use SCG-based approach. Due to this specific reason. So there is another reason why some cMAGs are not 100 % complete.

Reviewer #3 (Comments for the Author):

Eisenhofer et al. collected 22 mice fecal samples and evaluated the performance over four combinations of sequencing depth and technology in genome-resolved metagenomics. This study demonstrated the advantages of PacBio HiFi technology in metagenomics and contributed to the promotion of new sequencing technologies. Although this work provided a good metagenomics dataset including short reads and HiFi long reads, it seems to lack innovation compared with similar work. More exploration is needed to demonstrate the pros and cons of short reads, HiFi long reads and hybrid strategies in genome-resolved metagenomics.

(1) Compared with the existing work below, this work seems to provide only incremental results. Please describe the existing work in detail in the Introduction section and explain the necessity and urgency of this work.

Bickhart D M, Kolmogorov M, Tseng E, et al. Generating lineage-resolved, complete metagenome-assembled genomes from complex microbial communities[J]. Nature Biotechnology, 2022, 40(5): 711-719.

Kim C Y, Ma J, Lee I. HiFi metagenomic sequencing enables assembly of accurate and complete genomes from human gut microbiota[J]. Nature Communications, 2022, 13(1): 6367.

Jia L, Wu Y, Dong Y, et al. A survey on computational strategies for genome-resolved gut metagenomics[J]. Briefings in Bioinformatics, 2023: bbad162.

Orellana L H, Krüger K, Sidhu C, et al. Comparing genomes recovered from time-series metagenomes using long-and short-read sequencing technologies[J]. *Microbiome*, 2023, 11(1): 105.

(2) Compared with the mice gut representative microbial genome catalog such as mGMB (Liu C et al. *Nature Communications*, 2020), please check whether the HiFi long-read and hybrid strategies can recover higher quality or novel MAGs.

(3) When analyzing MAGs from long reads, it is difficult to show the advantages of long reads by only considering the completeness and contamination of CheckM. According to the Minimum information about a metagenome-assembled genome (MIMAG) by the Genomic Standards Consortium (GSC), high-quality MAG indicates that a MAG is >90% complete with less than 5% contamination and encodes the 23S, 16S, and 5S rRNA genes, and at least 18 tRNA genes. A comparative analysis of high-quality MAGs from short and/or long reads is essential.

Eisenhofer et al. collected 22 mice fecal samples and evaluated the performance over four combinations of sequencing depth and technology in genome-resolved metagenomics. This study demonstrated the advantages of PacBio HiFi technology in metagenomics and contributed to the promotion of new sequencing technologies. Although this work provided a good metagenomics dataset including short reads and HiFi long reads, it seems to lack innovation compared with similar work. More exploration is needed to demonstrate the pros and cons of short reads, HiFi long reads and hybrid strategies in genome-resolved metagenomics.

(1) Compared with the existing work below, this work seems to provide only incremental results. Please describe the existing work in detail in the Introduction section and explain the necessity and urgency of this work.

- ◆ Bickhart D M, Kolmogorov M, Tseng E, et al. Generating lineage-resolved, complete metagenome-assembled genomes from complex microbial communities[J]. *Nature Biotechnology*, 2022, 40(5): 711-719.
- ◆ Kim C Y, Ma J, Lee I. HiFi metagenomic sequencing enables assembly of accurate and complete genomes from human gut microbiota[J]. *Nature Communications*, 2022, 13(1): 6367.
- ◆ Jia L, Wu Y, Dong Y, et al. A survey on computational strategies for genome-resolved gut metagenomics[J]. *Briefings in Bioinformatics*, 2023: bbad162.
- ◆ Orellana L H, Krüger K, Sidhu C, et al. Comparing genomes recovered from time-series metagenomes using long-and short-read sequencing technologies[J]. *Microbiome*, 2023, 11(1): 105.

(2) Compared with the mice gut representative microbial genome catalog such as mGMB (Liu C et al. *Nature Communications*, 2020), please check whether the HiFi long-read and hybrid strategies can recover higher quality or novel MAGs.

(3) When analyzing MAGs from long reads, it is difficult to show the advantages of long reads by only considering the completeness and contamination of CheckM. According to the Minimum information about a metagenome-assembled genome (MIMAG) by the Genomic Standards Consortium (GSC), high-quality MAG indicates that a MAG is >90% complete with less than 5% contamination and encodes the 23S, 16S, and 5S rRNA genes, and at least 18 tRNA genes. A comparative analysis of high-quality MAGs from short and/or long reads is essential.

Dear Prof. Wei-Hua Chen and reviewers

First of all, thank you for the constructive and valuable comments and feedback. We have provided both a clean and marked-up version of the revised manuscript, and corresponding line numbers are indicated in this document. Below, we carefully address each of the comments and corrections provided by the reviewers.

Reviewer #1 (Comments for the Author):

Summary:

The authors compared seven performance metrics over four sequencing combinations from a laboratory mouse study. While different strategies were better at different aspect, the long-read gave the best assembly statistics and the hybrid approach gave the longest assemblies and highest percent of reads mapped back to genomes. This study thoughtfully explores the pros and cons of each of the four sequencing depth and technologies.

Minor Comments:

Line 127: "Binning and refinement were performed as above." Is not needed.

Fixed.

Line 157: There were no record found under ENA accession: PRJEB65885

This link should now be publicly accessible.

Major Comments:

There are no major comments; this has been a long-awaited comparison.

Reviewer #2 (Comments for the Author):

In the submitted manuscript Eisenhower et al. compare different sequencing strategies in the light of genome-resolved metagenomics. The manuscript is very well written and the results are clearly presented. I have to say that I really enjoyed reading this manuscript and I think the output from this manuscript is important to the field. Of course I have some comments and criticisms and they are listed below.

1. My main concern is related to binning. It's not clearly written (and it should be) how the original samples have been mapped back to the co-assemblies. I would assume that in the case of short-reads the authors have used all samples separately and mapped them. In long-read approach the original samples were pooled before sequencing, so there is only one (?) sample mapped back to the assembly. As most of the binning algorithms rely on nucleotide composition and differential coverage,

the information from a single sample mapping is very weak compared to differential coverage coming from 11 samples. So if this was the case, the differences seeing in the recovery of MAGs are actually stemming from the binning, not really from the sequencing strategy. I hope the authors would clarify this part and in case there was really different strategy in the binning between the approaches, this should be fixed to make the comparison really about sequencing strategy. And of course the binning software were totally different for the two short- and long-read approaches, I would assume this also affects the results. I don't think there would be any issues using the same approach for all strategies, but correct me if I'm wrong.

We did indeed coassemble the short read samples (specified in the first line of the second paragraph under the 'Data Analysis' subheader; LN 128 of tracked changes file). We also further clarified that long read assemblies were binned individually (LN 134 of tracked changes file). The reviewer is correct that differential coverage is indeed beneficial for binning, and that we were unable to use this information for the long-read binning. However, we don't see this being a major issue due to a couple of reasons:

- 1) The longer contigs provide more accurate nucleotide composition estimations, which help binning accuracy
- 2) We see the scenario where only a small number of samples are sequenced using long reads to be more realistic, due to the current costs of long read sequencing

We did indeed use a different binning approach for the short and long read approaches, but this was because others have noticed that better performance can be gained by binning this data differently from short reads. For example, the HiFi MAG pipeline is 'completeness-aware', preventing the improper binning of complete contigs. Additionally, SemiBin2 has been designed to handle long read data. More information on this, and a benchmark by Daniel Portik on multiple public datasets can be found here:

<https://github.com/PacificBiosciences/pb-metagenomics-tools/blob/master/docs/Tutorial-HiFi-MAG-Pipeline.md>

2. As this is a very methods oriented paper, the methods could be written with more details. It is not clear whether the authors used a different mapping strategy when mapping the short- and long- reads back to the assembly. Or did they use bowtie2 for both approaches when removing the host sequences? I think bowtie2 does not work well with long-reads. Again, correct me if I'm wrong.

We apologise for the confusion here, and have further clarified this in the methods (LN 123 & 135 of tracked changes file).

3. I would like to see some discussion on why the long-read approach yielded the largest assembly, but had a lower mapping rate compared to the other approaches. I would expect that a bigger assembly would also cover more of the original data. Indeed, one explanation is that while the long read assemblies were indeed larger, they were more redundant than the short read assemblies. We have added discussion of this at the end of the 'Quantity of recovered MAGs' section (LN 207-213 of tracked changes file).

4. Line 204: CheckM2 does not use SCG-based approach. Due to this specific reason. So there is another reason why some cMAGs are not 100 % complete. The reviewer is correct that CheckM2 does not use a SCG-based approach. Instead, it uses a machine learning model trained on ~5,000 complete isolate genomes. While the method is different, the rationale to explain why cMAGs are not estimated to be 100% complete is the same: the model has a level of uncertainty, as described by the authors in CheckM2 publication: (<https://www.nature.com/articles/s41592-023-01940-w>). We have added a line clarifying that CheckM2 estimates contain a level of uncertainty (LN 236 of tracked changes file).

Reviewer #3 (Comments for the Author):

Eisenhofer et al. collected 22 mice fecal samples and evaluated the performance over four combinations of sequencing depth and technology in genome-resolved metagenomics. This study demonstrated the advantages of PacBio HiFi technology in metagenomics and contributed to the promotion of new sequencing technologies. Although this work provided a good metagenomics dataset including short reads and HiFi long reads, it seems to lack innovation compared with similar work. More exploration is needed to demonstrate the pros and cons of short reads, HiFi long reads and hybrid strategies in genome-resolved metagenomics.

(1) Compared with the existing work below, this work seems to provide only incremental results. Please describe the existing work in detail in the Introduction section and explain the necessity and urgency of this work.

Bickhart D M, Kolmogorov M, Tseng E, et al. Generating lineage-resolved, complete metagenome-assembled genomes from complex microbial communities[J]. *Nature Biotechnology*, 2022, 40(5): 711-719.

Kim C Y, Ma J, Lee I. HiFi metagenomic sequencing enables assembly of accurate and complete genomes from human gut microbiota[J]. *Nature Communications*, 2022, 13(1): 6367.

Jia L, Wu Y, Dong Y, et al. A survey on computational strategies for genome-resolved gut metagenomics[J]. *Briefings in Bioinformatics*, 2023: bbad162.

Orellana L H, Krüger K, Sidhu C, et al. Comparing genomes recovered from time-series metagenomes using long-and short-read sequencing technologies[J]. *Microbiome*, 2023, 11(1): 105.

We agree with the reviewer that the present study is incremental in nature—it was delayed, and other papers directly comparing HiFi reads to short-reads came out before it was submitted. However, we don't see this as a major issue, as providing the field with more information regarding the pros and cons of HiFi sequencing is beneficial for researchers who are considering using it. We also foresee the data in our study being useful for future method development and benchmarking.

We already cite Bickhart et al. and Kim et al. in our introduction, though direct comparisons to the present work are not suitable, as Bickhart et al. used Hi-C binning, and Kim et al. did not compare HiFi to short-read assembly/binning. We have added the Orellana et al. reference and compared their findings (which are in agreement with ours) to the revised version. We have also added and discussed Gehrig et al. 2022 (<https://www.ncbi.nlm.nih.gov/pmc/articles/PMC9176275/>) and Zhang et al. 2023 (<https://doi.org/10.1093/bib/bbad087>). See updated text from LN 54 in tracked changes file.

(2) Compared with the mice gut representative microbial genome catalog such as mGMB (Liu C et al. *Nature Communications*, 2020), please check whether the HiFi long-read and hybrid strategies can recover higher quality or novel MAGs.

A comparison of our long-read / hybrid MAGs to a mice gut genome catalog (thousands of MAGs) would be too exhaustive, and we don't think it would add much to the paper (MAGs recovered using long-reads have already been shown to be of higher quality compared to short-reads; both in the present study and others).

However, we agree with the reviewer that some comparison of our data to a mice gut genome catalog would be beneficial for the reader. To this end, we compared the mapping rates of the mice samples in this study to the Mouse Gastrointestinal Bacteria Catalogue (MGBC; <https://doi.org/10.1016/j.chom.2021.12.003>), to see how a public MAG catalogue would compare to de-novo generated MAGs. Briefly, we dereplicated 26,640 MAGs using dRep at 98% ANI into 1363 MAGs. We then mapped the short-reads to this combined catalogue, which resulted in an average of 85-87% of reads being recruited. This number is similar to what we got for the de novo short-read MAGs in the present study (87-89%), and lower than what was obtained with the hybrid approach (93-94%). This analysis has been added to the revised manuscript (LN 215-228), and code has been uploaded to the GitHub repository: https://github.com/EisenRa/2023_HiFi_comparison_mice/blob/main/code/BASH_code.sh

(3) When analyzing MAGs from long reads, it is difficult to show the advantages of long reads by only considering the completeness and contamination of CheckM. According to the Minimum information about a metagenome-assembled genome (MIMAG) by the Genomic Standards Consortium (GSC), high-quality MAG indicates that a MAG is >90% complete with less than 5% contamination and encodes the 23S, 16S, and 5S rRNA genes, and at least 18 tRNA genes. A comparative analysis of high-quality MAGs from short and/or long reads is essential.

We have performed this comparison as suggested by the reviewer. Briefly, we used barrnap to annotate 5S, 16S, and 23S, and tRNA-scan to annotate tRNAs from the MAGs. Long reads produced 50 'high-quality' MAGs (MIMAG), hybrid produced 40, short-read 40Gbp produced 6, and 20Gbp produced 5. We have added these results to the updated manuscript (LN 239-244 of tracked changes file).

We wish to thank the reviewers again for taking the time to review our manuscript and for providing us with constructive feedback.

Re: Spectrum03590-23R1 (A comparison of short-read, HiFi long-read, and hybrid strategies for genome-resolved metagenomics)

Dear Dr. Raphael Eisenhofer:

Congratulations! Your manuscript has been accepted, and I am forwarding it to the ASM production staff for publication. Your paper will first be checked to make sure all elements meet the technical requirements. ASM staff will contact you if anything needs to be revised before copyediting and production can begin. Otherwise, you will be notified when your proofs are ready to be viewed.

Sincerely,
Wei-Hua Chen
Editor
Microbiology Spectrum

Reviewer #2 (Comments for the Author):

I have no further comments.

Reviewer #3 (Comments for the Author):

All issues have been resolved.